# Evaluation of In Vitro Distribution and Plasma Protein Binding of Selected Antiviral Drugs (Favipiravir, Molnupiravir and Imatinib) against SARS-CoV-2

**DOI:** 10.3390/ijms24032849

**Published:** 2023-02-02

**Authors:** Orsolya Dömötör, Éva A. Enyedy

**Affiliations:** MTA-SZTE Lendület Functional Metal Complexes Research Group, Department of Inorganic and Analytical Chemistry, Interdisciplinary Excellence Centre, University of Szeged, Dóm tér 7, H-6720 Szeged, Hungary

**Keywords:** tyrosine kinase inhibitor, COVID-19, proton dissociation, lipophilicity, plasma protein binding, binding constants

## Abstract

There are a number of uncertainties regarding plasma protein binding and blood distribution of the active drugs favipiravir (FAVI), molnupiravir (MOLNU) and imatinib (IMA), which were recently proposed as therapeutics for the treatment of COVID-19 disease. Therefore, proton dissociation processes, solubility, lipophilicity, and serum protein binding of these three substances were investigated in detail. The drugs display various degrees of lipophilicity at gastric (pH 2.0) and blood pH (pH 7.4). The determined p*K*_a_ values explain well the changes in lipophilic character of the respective compounds. The serum protein binding was studied by membrane ultrafiltration, frontal analysis capillary electrophoresis, steady-state fluorometry, and fluorescence anisotropy techniques. The studies revealed that the ester bond in MOLNU is hydrolyzed by protein constituents of blood serum. Molnupiravir and its hydrolyzed form do not bind considerably to blood proteins. Likewise, FAVI does not bind to human serum albumin (HSA) and α1-acid glycoprotein (AGP) and shows relatively weak binding to the protein fraction of whole blood serum. Imatinib binds to AGP with high affinity (log*K*′ = 5.8–6.0), while its binding to HSA is much weaker (log*K*′ ≤ 4.0). The computed constants were used to model the distribution of IMA in blood plasma under physiological and ‘acute-phase’ conditions as well.

## 1. Introduction

The outbreak of the COVID-19 pandemic in the first months of 2020 posed a serious challenge for medical establishments worldwide. Rapid spread and the unpredictable outcome of the infection required the urgent development of effective drug therapies to prevent the progression of severe illness and to stop transmission of SARS-CoV-2. In this respect, off-label use of drugs previously approved for other illnesses emerged as a key strategy in the fight against the virus. Instead of the slow and costly development and approval of novel drugs, physicians started the clinical testing of antiviral agents originally developed against other types of RNA viruses like hepatitis C, HIV, or influenza. Remdesivir, ribavirin, lopinavir, favipiravir (FAVI), and molnupiravir (MOLNU) have been used to treat mild to severe COVID-19 cases in several countries based on such considerations [1,2,3,4]. Both FAVI and MOLNU (Figure 1) are nucleoside analog prodrugs that were originally designed for treatment of influenza disease; FAVI was approved for this application in Japan in this indication [5,6,7]. Favipiravir is metabolized in cells to an active form, favipiravir-ribofuranosyl-5’-triphosphate, and inhibits viral RNA polymerase [6]. Molnupiravir acts in a slightly different way, namely, the isobutyrate function is hydrolyzed during absorption to form N(4)-hydroxycytidine (EIDD-1931), which is then triphosphorylated at the 5′-end in the cellular environment and promotes mutations in viral RNA replication [1,8,9,10]. Favipiravir was authorized for treatment of COVID-19 in Japan, Russia, Serbia, Turkey, India, Egypt, and Hungary etc. under emergency provisions, while MOLNU was authorized in the US, UK, Bangladesh, and Israel. In addition to antiviral agents, drugs that modulate the extreme immune response triggered by infection have attracted attention as well. This group includes, among others, corticosteroids, interferon-1, thalidomide, (hydroxy)chloroquine and various cytokine and kinase inhibitors [3,4]. In this regard, the tyrosine kinase inhibitor (TKI) imatinib (IMA) (Figure 1), originally designed for the treatment of chronic myeloid leukaemia (CML), has also been proposed as a potential therapy for COVID-19 disease [11,12]. In CML patients treated with TKIs, extremely low incidence of COVID-19 infection was reported, and in vitro data revealed that IMA as a Bcr-Abl tyrosine kinase inhibitor may block viral replication in the early stages of infection [11,13]. The immunomodulatory effect of IMA was also claimed to be capable of reducing severe inflammatory response [12].

All three medicines are delivered orally, otherwise their pharmacokinetic behavior differs significantly. Simple physicochemical features such as solubility, charge, and lipophilicity, as well as plasma protein binding are extremely important in terms of drug absorption, distribution, elimination, toxicity, and efficacy. Interestingly, there is little information available on these qualities in the cases of FAVI and MOLNU, and predicted data found in the common databases (DrugBank, PubChem) are often contradicting [14,15,16,17]. The product monograph for Avigan^®^ (FAVI) reports good aqueous solubility and 54% serum protein binding of the drug observed in vitro [14]. This in vitro protein binding data appears to be referred to in the literature [17,18,19,20,21] as an in vivo result [17,18,19,20,21]. The fluorescence of FAVI was reported only recently in 2021 by S.M. Megahed et al. [22]. At the same time, several case reports on the fluorescence of nails, hair, skin, face, and sclera of patients were published by Turkish clinicians after treatment with FAVI, although this phenomenon was associated with the active ingredient in only one publication, using Wood’s light for examination of patients and the halved tablets [23,24,25]. No plasma protein binding was observed for EIDD-1931 in vitro, and no such data is available for MOLNU [15]. The interaction of IMA with human serum albumin (HSA) and α1-acid glycoprotein (orosomucoid, AGP) has been reported in the literature, but the role of these two proteins has not been clearly elucidated yet [26,27,28,29,30,31,32,33]. Both HSA and AGP are acute phase proteins (APP), namely their concentrations change considerably in the blood in response to inflammation. The serum concentration of AGP increases two- to three-fold during acute phase response (positive APP), whereas HSA concentration decreases during inflammation (negative APP) [34,35,36]. The orosomucoid AGP is the second most important transport protein for organic drugs in blood, next to HSA. At the same time, physiological concentration of AGP is much lower (approximately 10–20 μM) in comparison to albumin (ca. 630 μM) [36,37]. Therefore, serum protein binding of IMA can be severely affected by inflammation, which is a possible consequence of viral infections or tumor diseases [32,38,39]. Neutral or negatively charged drugs bind primarily to HSA, while drugs possessing basic functional groups tend to show affinity for AGP [36,37,40]. Taking this observation into account, different protein binding preferences are expected for FAVI, MOLNU and IMA in blood plasma.

The three orally administered heterocyclic aromatic drugs with dissociable moieties, FAVI, MOLNU and IMA, were selected here for detailed in vitro blood distribution studies. The available literature on these compounds’ protonation processes, lipophilicity, thermodynamic aqueous solubility, and blood serum protein binding is fairly limited. Our goal was to provide a comprehensive overview of these properties as well as to critically examine the literature data and potential shortcomings of the approaches used in this work. The proton dissociation processes were monitored by pH-potentiometry, UV–visible and fluorescence spectroscopies, and/or by *n*-octanol/water partition experiments also considering the solubility and stability issues of the compounds under investigation. The interaction of the compounds with whole blood serum, and the binding to serum proteins HSA and AGP was investigated as well. Results on the protein binding were obtained using a combination of separation techniques such as membrane ultrafiltration and frontal analysis capillary electrophoresis (FACE), as well as spectroscopic methods, ^1^H NMR spectroscopy, steady-state fluorometry and fluorescence anisotropy.

## 2. Results and Discussion

### 2.1. Aqueous Stability and Solubility

Molnupiravir, one of the title compounds, contains an ester functional group that may hydrolyze under certain conditions. Therefore, the aqueous stability of MOLNU was investigated as a first step at pH 2.0, 7.4, and 12.0 by ^1^H NMR spectroscopy. The NMR spectra (Figure 1) show that MOLNU hydrolyzes completely within 30 min at pH 12.0 resulting in isobutyrate and N(4)-hydroxycytidine (EIDD-1931). This process is much slower at pH both 2.0 and 7.4 than at pH 12; and in 5 days only 8% and 2% hydrolyzed products, respectively, were obtained. Deamination, which is characteristic for cytidine under alkaline and acidic conditions, was not observed probably due to the replacement of amine by a hydroxylamine group in the molecule [41,42]. The other two compounds, FAVI and IMA were stable in aqueous media at pH 2 and 7.4 at least for 30 h (Appendix A).

The studied compounds (Figure 1) display different thermodynamic solubility (*S*) at various pH values and temperatures. Favipiravir and MOLNU exhibit excellent solubility (10 mM solutions could be prepared) at pH 2.0 and 7.4, both at 25 °C and 37 °C, while the solubility of IMA decreases with increasing pH and temperature (Figure 2). As a result of the gradual deprotonation of the charged groups, the good solubility attained at pH 2.0 (*S* > 10 mM) declines to 14 µM by pH 8.2 (*t* = 37 °C), where the neutral form predominates (vide infra).

### 2.2. Proton Dissociation Processes

Table 1 summarizes the proton dissociation constants (p*K*a) measured using pH-potentiometric, UV-visible (UV-vis) spectroscopic, and fluorometric titrations. *n*-Octanol/water partition experiments were used as well in the case of IMA and FAVI to obtain p*K*_a_ values.

One p*K*_a_ value could be determined for FAVI by various techniques which were in good agreement with each other. The *n*-octanol/water distribution coefficients (*D*_pH_) obtained at various pH values reveal the more hydrophilic character of the deprotonated form (Figure 3), as deprotonation of the 3-hydroxy-pyrazine ring results in a negatively charged ion. In solution, FAVI can exist in keto-enol tautomeric forms, although the enol form was reported to be more stable in aqueous solution [43], thus the proton is likely to dissociate from the enol group or the iminol/amide group (Appendix A). Interestingly, based on theoretical calculations only protonation of the neutral molecule was taken into account, and proton dissociation was not investigated [43,44]. As a result of deprotonation, the weak fluorescence intensity of FAVI increases about ten-fold and bathochromic shift in the UV-vis absorption spectrum occurs as well (Figure 4 and Appendix A). This spectral shift is indicative of the formation of a more extended π-conjugated system suggesting the formation of the enolate tautomer upon deprotonation (Appendix A).

Two p*K*_a_ values could be determined for MOLNU by pH-potentiometric and spectrophotometric titrations, which belong to the N(4)-hydroxycytosine moiety. The UV-vis spectra in Figure 5 show characteristic changes between pH 2.0–4.0 and 9.5–11.5, which are not sensitive to the hydrolysis of the ester function. Repeated pH-potentiometric titration of the hydrolyzed MOLNU resulted in the appearance of an additional p*K*_a_ = 4.54 ± 0.08, that corresponds to the reported p*K*_a_ of isobutyric acid (p*K*_a_ = 4.67, *I* = 0.1 M NaNO_3_) [45], while the original p*K*_a_ values were barely affected (see titration curves in Appendix A). The fully protonated form is positively charged (H_2_L^+^), and the molecule becomes neutral by the first proton dissociation process. Given the trend of lipophilicity data determined at pH 2.0 and 7.4, the zwitterionic character of this molecule is not plausible (vide infra). The second deprotonation occurs in the alkaline pH range (p*K*_a_ = 10.18 ± 0.02 determined by pH-potentiometry). The widely studied analogue, cytidine is characterized by one p*K*_a_ = 4.1–4.5 [46,47] assigned to the N(3)H^+^ group. The p*K*_a1_ = 2.14 ± 0.01 of MOLNU is assumed to belong to the same group. The second deprotonation possibly takes place on the hydroxyamino/hydroxyimino functional group. The constants determined are in good agreement with the data in the product assessment report of the European Medicines Agency (p*K*_a1_ = 2.2, p*K*_a2_ = 10.2), however the existence of a third deprotonation, found in the same report (p*K*_a3_ = 12) [15], was not observed by us in the studied pH range.

Imatinib possesses four basic groups which can be protonated (Figure 1, Table 1). The lowest p*K*_a_ determined by UV-vis titration corresponds to the secondary ammonium group. This value, together with p*K*_a2_ and p*K*_a3_ determined by pH-potentiometric titrations, agrees very well with the reported data of Szakács et al. (*I* = 0.15 M NaCl) [48]. The latter two are macroscopic constants belonging to the pyridinium and one of the piperazinium nitrogens. Imatinib becomes neutral by dissociation of the last proton from the remaining piperazinium group and its solubility drops to the low-micromolar range (see Figure 2), making pH-potentiometric determination difficult. Additionally, this process is not associated with any spectral change in the UV-vis spectrum. Therefore, *n*-octanol/water distribution coefficients were determined at various pH values (Appendix A) and p*K*_a4_ = 7.9 ± 0.1 could be computed, which is two tenths higher than the value reported by Szakács et al. (determined by pH-potentiometry using the titration curve until no precipitation occurred). Given the uncertainty of both methods, the agreement of these constants is acceptable.

The knowledge of these dissociation constants allowed us to compute fractions of the actual protonation states at various pH values (Appendix A) including pH 7.4 (pH of extra- and intracellular space) and pH 2.0 (a typical gastric pH) (Table 1). At pH 7.4, MOLNU has the neutral HL form, whereas FAVI is entirely deprotonated (L-), and IMA is mostly positively charged, with just around 25% neutral form. On the other hand, at pH 2.0 the protonated forms H_4_L^4+^ and H_3_L^3+^ predominate for IMA, FAVI is neutral (100% HL), whereas 58% and 42% of MOLNU are present in H_2_L^+^ and HL forms, respectively.

**Table 1 ijms-24-02849-t001:** Proton dissociation constants (p*K*_a_) of the studied compounds determined by pH-potentiometry (pHm) UV-vis, fluorometry (fluor) or *n*-octanol/water partition (o/w part.), *n*-octanol/water distribution coefficients (log*D*_pH_), and calculated distribution (%) of the species in different protonation states at pH 7.40 and 2.0 [*I* = 0.1 M (KCl); *t* = 25 °C].

	IMA	FAVI	MOLNU
Proton dissociation processes			
	p*K*_a1_ 1.72 ± 0.01 (UV-vis) p*K*_a2_ 3.06 ± 0.05 (pHm) p*K*_a3_ 3.86 ± 0.03 (pHm) p*K*_a4_ >7.0 (pHm) 7.9 ± 0.1 (o/w part.)	p*K*_a1_ 5.11 ± 0.01 (pHm) 5.08 ± 0.01 (UV-vis) 5.17 ± 0.01 (fluor) 5.1 ± 0.1 (o/w part.)	p*K*_a1_ 2.14 ± 0.01 (pHm) 2.11 ± 0.01 (UV-vis) p*K*_a2_ 10.18 ± 0.02 (pHm) 10.34 ± 0.01 (UV-vis)
n-octanol/water distribution			
log*D*_2.0_ 25 °C	−2.6 ± 0.01	+0.15 ± 0.01	−0.68 ± 0.02 ^a^
log*D*_7.4_ 25 °C	+2.4 ± 0.1	−1.99 ± 0.06	−0.29 ± 0.02 ^a^
log*D*_7.4_ 37 °C	+2.9 ± 0.1	−1.82 ± 0.05	−0.17 ± 0.03
Species at pH 7.4			
	HL^+^: 76% ^b^ L: 24% ^c^	L^−^:100%	HL: 100%
Species at pH 2.0			
	H_4_L^4+^: 33% H_3_L^3+^: 62%	HL:100%	H_2_L^+^: 58% HL: 42%
	H_2_L^2+^: 5%		

^a^ Lipophilicity data for the hydrolyzed products h-MOLNU at 25 °C: log*D*_2.0_ = −1.25 ± 0.04, log*D*_7.4_ = −2.00 ± 0.05; ^b^ HL^+^: 67%, L: 33% using p*K*_a4_ = 7.7 determined by Szakács et al. [48].

### 2.3. Lipophilicity

Lipophilicity of the studied drugs has been scarcely reported in the literature [15]. Table 1 comprises the logarithm of distribution coefficients (log*D*_pH_) determined at pH 2.0 and 7.4. Imatinib appears to be highly hydrophilic, more than MOLNU, at pH 2.0; however, gradual loss of positive charges provides a rather lipophilic molecule at pH 7.4. Molnupiravir retains its hydrophilic character even at this pH thanks to the polar sugar moiety. Hydrolyzed MOLNU (h-MOLNU, prepared by alkaline hydrolysis of MOLNU) was also studied and it is somewhat more hydrophilic than the parent molecule. Favipiravir was proven to be amphiphilic at pH 2.0 and it is mainly hydrophilic at pH 7.4 where the deprotonated form is predominant. The *n*-octanol/water distribution of the compounds at pH 7.4 was assayed at 37 °C as well, the coefficients move slightly to the (more) lipophilic range at this temperature.

### 2.4. Fluorescence Properties of FAVI and IMA

Among the three studied antiviral agents, only FAVI exerts measurable fluorescence emission in aqueous solution as it was shown in the former section (Figure 4). This phenomenon was further investigated in various solvents and fluorescence lifetime parameters were also determined by time correlated single photon counting (TCSPC) (Table 2, Appendix A). Favipiravir displays the most intense fluorescence when it is deprotonated. A single excitation peak is observed, which corresponds well to the absorption maximum (λ = 362 nm). The neutral form dissolved in water, ethanol or *n*-octanol shows much smaller fluorescence intensity, and the two bands in the excitation spectra reflect the shape of the absorption spectra recorded in the same solvents. As it is known that planar aromatic structures are the most fluorescent, the aromatic character of the deprotonated form can be assumed. In the case of the neutral form, the retained but much lower fluorescence and the dual absorption and excitation bands refer to the parallel existence of more tautomers under this condition, from which one form preserves the aromatic structure. Fluorescence lifetime data confirm the presence of two kinds of fluorophores in acidic aqueous, ethanolic and *n*-octanolic solutions of FAVI. The minor species possess shorter lifetime (*τ*_1_ = 3.8–6.4 ns), while the lifetimes of the longer-lived forms (*τ*_2_ = 8.05–9.74 ns) are rather similar to that of the deprotonated form (*τ* = 10.00 ns). Our findings suggest that FAVI (or its fluorescent metabolites) is most likely responsible for the fluorescence of nails, hair, skin, face, and sclera of FAVI-treated patients [23,24,25].

In both ethanol and *n*-octanol, IMA and MOLNU (and h-MOLNU) fluorescence was minimal in aqueous solution at any protonation state. In *n*-hexane, IMA has weak fluorescence (λ_EX_ = 287nm, λ_EM_ = 450 nm, Appendix A), whereas MOLNU is not in this solvent.

### 2.5. Interaction with Blood Serum Proteins Human Serum Albumin and α1-Acid Glycoprotein

Our experiments on global serum protein binding and interaction with HSA and AGP are reported in detail in this section, and the acquired results are discussed and compared with literature data in Section 2.6. The diverse aqueous solubility of the compounds required a complex methodological approach to study their serum protein binding. Consequently, this involved the combined use of spectrofluorometry, ^1^H NMR spectroscopy, ultrafiltration–UV-vis and capillary electrophoresis frontal analysis techniques.

In the first step, the interaction of the three medicines (at 50 μM concentration) with whole human serum was investigated using ultrafiltration. Following the filtration, the low molecular mass (LMM) fraction of the samples was studied by UV-vis spectrophotometry. As depicted in Figure 6, the compounds exhibit diverse behavior. By filtering the compounds dissolved in PBS (dashed spectra), IMA stuck to the filter to a high extent (85%) most likely due to its limited solubility at pH 7.40. Molnupiravir and FAVI could be recovered in 78% and 100% after filtration, respectively. Then compounds were ultrafiltered in the presence of 4-fold diluted blood serum and the adhesion was considered. Practically no free IMA was detected in solution, FAVI is bound in 18 ± 2% and MOLNU showed no measurable binding to the serum protein fraction. However, the hydrolytic state of MOLNU is not known in this experiment as the hydrolyzed forms (EIDD-1931 and isobutyrate) have the same absorption spectrum as the parent compound. Therefore, the stability of MOLNU was investigated in the presence of blood serum by ^1^H NMR spectroscopy. MOLNU at 1 mM final concentration was mixed with 2-fold diluted blood serum and the reaction was followed in time as it is shown in Figure 7. The colored frames show well that a new set of peaks appears in the spectra that can be attributed to isobutyrate and EIDD-1931. The hydrolysis is relatively slow and was completed in more than one day (Appendix A). The pseudo-esterase activity of HSA is known from the literature [37,49,50]; furthermore, divalent metal ions such as Cu(II) ion may induce ester hydrolysis as well [51]. Therefore, as a next step, the effect of HSA (630 μM) and the LMM fraction of blood serum, possibly containing traces of Cu(II) ion, was investigated in separate experiments. Neither of the two media could induce hydrolysis of MOLNU within two days. The same behavior was observed in the presence of 15 μM AGP or 45 μM human transferrin. Knowing that orally taken MOLNU does not practically reach the circulatory system, but h-MOLNU does, the serum protein binding studies were done with the hydrolyzed form, which are supplemented with the results obtained for MOLNU for comparison.

The serum protein binding of h-MOLNU (30 μM) was investigated in ultrafiltration experiments and no binding to HMM components of blood serum or binding to the single proteins HSA (630 μM) or AGP (15 and 30 μM) was observed (Appendix A). Interestingly, MOLNU was bound to AGP in 4 ± 1% and 8 ± 1% in the presence of 0.5 and 1 equiv. AGP, respectively, while ultrafiltered samples of blood serum indicated less than 2% binding to the HMM fraction (Appendix A). Moreover, for FAVI, less than 2% of the bound fraction of the compound could be detected in ultrafiltration experiments by the addition of different concentrations of HSA or AGP (see Appendix A). At the same time, moderate binding to HMM components (ranging from 18 to 28%) was detected when FAVI was dissolved in (diluted or non-diluted) whole blood serum. The highest (28%) serum protein binding of FAVI was found under the conditions being most similar to the physiological ones (200 μM FAVI in non-diluted blood serum). When 50 μM IMA and equimolar HSA was ultrafiltrated, about 40% remained in the HMM fraction (Appendix A), however these data should be handled cautiously, due to the high adhesion tendency of the compound to the filter surface.

Due to the adhesion phenomenon of IMA to the ultrafiltration membrane, frontal analysis capillary electrophoresis (FACE) was utilized to investigate the interaction of the drug with HSA and AGP. The FACE technique enables the study of rapid equilibrium processes unlike capillary zone electrophoresis. Herein, a large volume of pre-equilibrated sample was introduced into the capillary, and overlapping plateaus appear in the electroferogram instead of well separated zonal peaks. The concentration of the unbound drug can be determined from the plateau height of the free portion of the drug. The electrophoretic conditions were optimized for first, namely using PBS buffer as BGE and injection time of 30 s resulted in a reasonable plateau shapes and analysis time (Figure 8 and Appendix A). Imatinib is partly positively charged under this condition (Table 1), therefore it migrates before the negatively charged high molecular weight protein. In Figure 8, the effect of increasing amounts of AGP on the plateau height of free IMA can be followed. Gradually decreasing plateau heights are proportional to the free fraction of IMA and formation curve in Figure 8b for the AGP–IMA system can be calculated as well. The curve becomes saturated at an [IMA]_bound_/*c*_AGP_ ratio of close to 1, thus binding of maximum one IMA per AGP is feasible. The binding constant log*K*′ = 5.2 ± 0.1 was calculated with the PSEQUAD program [52] and the fitted binding curve is in good agreement with the experimental data points (Figure 8b). The addition of HSA instead of AGP affected barely the plateau height of IMA, and no considerable binding was found in the presence of up to 5 equiv. albumin. Rather similar behavior was seen in the case of HSA–FAVI and AGP–FAVI chemical systems in FACE studies (see Appendix A for representative electropherograms recorded for the HSA–FAVI system). This finding is consistent with the observations of ultrafiltration experiments (Appendix A).

The protein binding of the title compounds was further investigated by spectrofluorometric measurements. Albumin possesses three main hydrophobic drug binding pockets, referred to as Sudlow’s site I (in subdomain IIA), Sudlow’s site II (in IIIA) and site III (in IB). In these pockets, HSA binds and transports mostly lipophilic, neutral or negatively charged endogenous and exogenous compounds [37,53]. Binding at site I can be followed by fluorometry directly through the quenching of tryptophan-214 (Trp-214) residue. Moreover, fluorescent site markers such as warfarin (WF, for site I) and dansylglycine (DG, for site II) can be used for site specific binding assays [54,55]. In AGP, there are seven binding sites with varying affinities and capacities, however, all basic drugs, together with the acidic ones, can bind into the hydrophobic central cavity of AGP. For practical purposes, only this binding site may be considered for clinical relevance [36]. The Trp25 and Trp122 residues lie inside and next to the central binding cavity, respectively [40], therefore fluorescence quenching experiments are suitable to follow drug binding here. The displacement of dipyridamole (DIP) from AGP is another, less well-known approach to monitor the binding of a compound to AGP via fluorescence anisotropy measurements [56].

The three-dimensional fluorescence spectra in Figure 9 recorded for the AGP–IMA and HSA–IMA systems show well the differences between the binding affinities of the drug to the two proteins. The fluorescence of AGP (at λ_EX_: 280 nm, λ_EM_: 325 nm) reduced to ca. 60% upon addition of IMA and in turn the weak induced fluorescence of IMA (at λ_EX_: 280 nm, λ_EM_: 465 nm) appears in the spectrum. At the same time, light emission of HSA is barely affected by IMA and the induced fluorescence of the latter could not be observed. Favipiravir did not quench the fluorescence of HSA or AGP under the same conditions, and the presence of the proteins had no effect on the intense fluorescence band of FAVI (see Appendix A for 3D spectra).

Further titration experiments were carried out, to gain a deeper understanding of the binding interaction. Representative emission spectra obtained from the titration of the AGP–IMA system are presented in Figure 10. The fluorescence of AGP mainly originates from the three Trp amino acid residues. The binding of IMA effectively quenches this fluorescence and, in parallel, the induced emission band of IMA shapes a saturation curve (Figure 10). The position of the emission peak (λ_EM_ = 465 nm) together with the excitation maximum (λ_EX_ = 280 nm) of the bound IMA resemble to those obtained in *n*-hexane, verifying that the binding takes place in the hydrophobic cavity of the protein. The quenching studies were repeated at 37 °C, and the effect of long-term storage of frozen AGP stock solutions in plastic microtubes was investigated as well. The reason for the later experiment was, that plasticizers used for plastic containers and rubber wares can reportedly disrupt the binding of drugs to AGP [36,57]. All spectra were used for computing quenching constants shown in Table 3 with the computer program PSEQUAD [52], and a very good match was found between the experimental and calculated fluorescence intensities (see an example in Figure 10b). The determined values indicate that the binding affinity is somewhat higher at 37 °C in contrast to the constant obtained at room temperature (log*K*′ = 6.0 ± 0.1 vs. 5.8 ± 0.1, respectively). The effect of the long-term storage in plastic microtubes on the binding ability of AGP also appears to be moderate (Δlog*K*′ = 0.2).

Figure 11 shows little quenching when AGP was titrated with FAVI, h-MOLNU, or MOLNU. Even 91 equiv. of FAVI quenched in only 6% the fluorescence of AGP (Appendix A), and the emission band of FAVI itself (see Appendix A at 430 nm) is barely affected in the presence of this protein. Both h-MOLNU and MOLNU, similarly to FAVI, did not quench the fluorescence of AGP.

The binding of IMA, h-MOLNU and FAVI to AGP was also investigated in dipyridamole (DIP) displacement experiments. Dipyridamole is a drug utilized for decreasing platelet aggregation and for coronary vasodilatation. It binds to AGP at one site with extremely high affinity that is located in the hydrophobic pocket of the glycoprotein, and at least one additional low affinity site for DIP is assumed as well [58]. Dipyridamole is highly fluorescent in the visible range (λ_max(EX)_ = 400 nm, λ_max(EM)_ = 500 nm) and its steady-state fluorescence spectrum was not very sensitive to the binding on AGP (Appendix A). On the other hand, fluorescence anisotropy is an excellent method to follow the protein binding of DIP. Briefly, fluorescence anisotropy experiments provide information on the size and shape of a fluorophore via measurement of the polarized emission of a solution [59]. AGP-bound DIP possesses much higher anisotropy due to the slower rotational diffusion of the protein in comparison to free DIP (own data: *r*_bound_ ≈ 0.25; *r*_free_ ≈ 0.003; at pH = 7.40 (PBS), 25 °C). As Figure 12 shows, a considerable amount of DIP is bound to AGP (*r* = 0.14) under the applied conditions, and the anisotropy of the sample decreases by the addition of IMA in line with the increase of free and rapidly rotating fraction of DIP. It is noteworthy that h-MOLNU was not able to displace DIP from its binding pocket (Figure 12). The same experiment performed with FAVI is a good example to draw attention to the possible false interpretation of fluorescence anisotropy data where the hypothesized competitor itself (FAVI) displays fluorescence under the conditions used. A decrease in anisotropy is also observed here, however (according to the ultrafiltration, FACE, and fluorescence quenching experiments) no interaction between the studied species takes place (see details under Appendix A). Taking advantage of the intrinsic fluorescence of FAVI, direct measurements on its fluorescence anisotropy were carried out in the absence and presence of AGP. The anisotropy value did not increase considerably in the presence of 10 equiv. AGP (*r*_free FAVI_ = 1.8 × 10^−3^ ± 4 × 10^−4^, *r*_FAVI with AGP_ = 2.5 × 10^−3^ ± 5 × 10^−4^).

Binding events at the hydrophobic sites of HSA were investigated in Trp-214 quenching and site marker displacement experiments. Both WF and DG are high affinity fluorescent markers of hydrophobic sites I and II, respectively. The Trp-214 quenching also provides information on the binding at site I. The Trp-214 fluorescence was weakly quenched by the addition of IMA to HSA, and only 25% and 27% quenching was observed by the addition of 37 equiv. Imatinib at 25 °C and 37 °C, respectively (Figure 11b). Based on these data, the binding constants log*K*′ ≤ 4.0 were estimated. The WF displacement experiments displayed no replacement of bound WF, on the contrary a small, 17% increase of the fluorescence can be observed (Appendix A). This cannot be explained by the induced fluorescence of IMA, as it did not show any fluorescence by the addition of HSA (in contrast to AGP) (see Figure 9). Possibly the weak binding of IMA elsewhere results in somewhat different arrangement of interacting moieties in site I, e.g., more hydrophobic environment around bound WF. Imatinib was unable to displace DG from the hydrophobic site II of HSA (Appendix A). Thus, no binding constants could be calculated from the site marker displacement experiments. Similarly, FAVI, h-MOLNU and MOLNU did not significantly quench the fluorescence of Trp-214 (Figure 11 and Appendix A). The WF displacement studies also showed no competition of these compounds for site I. No measurement was applicable for the HSA–DG–FAVI ternary system due to the overlap of fluorescence bands of DG and FAVI. Based on our data, h-MOLNU and MOLNU could not displace DG from site II.

### 2.6. Discussion of the In Vitro Blood Serum Distribution of the Compounds

Studying the in vitro hydrolysis of MOLNU in blood serum appears to be a theoretical problem, because orally administered MOLNU is hydrolyzed by carboxylesterases during or after absorption to deliver EIDD-1931 into the systemic circulation [15]. At the same time, it may be important to monitor the fate of MOLNU following intravenous administration, where the drug directly enters the circulation. Our results showed relatively slow (ca. one day) hydrolysis of MOLNU in whole blood serum, which is clearly due to the presence of HMM components. The rate of the hydrolysis can be somewhat different at physiological conditions, where, in contrast to the experimental conditions 37 °C and low micromolar concentration of MOLNU applies. Human serum albumin is an obvious suspect, as its pseudo-esterase activity was demonstrated in numerous experiments with respect to acetylsalicylic acid, fatty acid esters, ketoprofen glucuronide, cyclophosphamide etc. [37,49,50]. Nevertheless, HSA could not hydrolyze the ester bond of MOLNU. Besides HSA, butyrylcholinesterase (pseudo-cholinesterase), and paraoxonase exerts esterase activity in blood plasma and traces of acetylcholinesterase (8 ng/mL) are present as well in blood [50]. The typical substrates of paraoxonase are likely phosphate esters, therefore butyrylcholinesterase, present in ca. 2.3–6.8 μg/mL in blood plasma, is more likely responsible for the enzymatic cleavage of the ester bond in MOLNU [50,60]. In fact, hydrolysis of MOLNU is likely to be faster in circulating blood due to the extensive esterase activity of the liver [61]. The present studies implemented by ultrafiltration and spectrofluorometric techniques indicated no considerable binding of h-MOLNU or MOLNU to serum proteins.

To the best of our knowledge, the binding of FAVI to human serum proteins was only investigated in vitro. The Japanese product monograph of Avigan^®^ tablets reports a 53–54% protein bound fraction of FAVI (*c*_FAVI_ = 1.7–173 μM); furthermore, the binding was attributed to HSA and AGP in 65% and 6.5%, respectively [14]. There is confusion in the literature, as the 54% overall protein binding of FAVI is commonly cited as in vivo data [17,18,19,20,21], although the adverted sources (if they are indicated) also seem to use this value as a non-referred literature data [20]. Moreover, this value is practically identical with the producer’s in vitro data (vide supra). Unfortunately, the monograph is not very detailed on the methodology: ultrafiltration was done, most probably by mixing of blood serum samples of various species (dog, rat, rabbit, human; obtained in advance) with FAVI; and it is not clear what kind of additional analytical method was applied to determine the bound HSA-to-AGP ratio [14]. The in vivo serum protein binding assays were performed only on rats and monkeys [14]. In vitro ultrafiltration experiments in the present study showed ca. 28% binding of FAVI to the HMM fraction of whole blood serum when 200 μM drug (corresponding to an average blood plasma level [14]) was interacted with non-diluted human blood serum. In addition, our ultrafiltration (Appendix A) and FACE studies implemented with single proteins HSA and AGP strongly suggest that neither of the two proteins are involved in binding of FAVI. Our spectrofluorometric assays also confirmed these observations. Discrepancies between the literature and present data may originate from the different methodology applied, or possible ethnic or regional differences should be taken into account in future studies, as it was noted in regards of the differing plasma levels of FAVI in Japanese patients and patients form the US [21].

Clinical relevance of serum protein binding of IMA is widely reported [29,33,38,57]. The role of AGP has been demonstrated in several in vivo and in vitro studies [32,33,38,39]. Elevated blood level of AGP was associated with delayed or lack of response to IMA treatment [33,57]. On the other hand, some studies reported low affinity or no binding of IMA to AGP [27,62]. These latter observations are possibly due to the non-physiological conditions applied (Table 3) [27] or to the pre-treatment of AGP used for the assays [62]. Moreover, it has long been reported that plasticizers can disrupt the binding of drugs to AGP; the impact of diethylhexyl phthalate (DEHP), released from PVC blood collection bags, on the free fraction of IMA in human plasma was also assessed [63]. Although microcentrifuge tubes are generally made of polypropylene, several water soluble leachables are found in them as typical processing agents and additives, including aromatic agents (e.g., 3,4-dimethylbenzaldehyde, Millad-3988) [64]. Our results showed only a little deviation between the computed binding constants when AGP stock solution was prepared freshly or stored for 2 months in ‘low-price’ plastic microtube at −20 °C (log*K*′ = 5.8 ± 0.1, 6.0 ± 0.1, respectively). The binding affinity was also somewhat higher when samples were thermostated at 37 °C (log*K*′ = 6.0 ± 0.1) instead of 25 °C; however, the difference, just like in the previous relationship, was not significant. These constants are somewhat lower than those reported by Fitos et al. and Gambacorti-Passerini et al., but fall within the same range and reveal high affinity binding of IMA on AGP [26,32]. The FACE experiments yielded an overall AGP–IMA binding constant (log*K*′ = 5.2 ± 0.1) that was approximately four times lower than the spectrofluorometric quenching data. Similarly, no binding could be observed when the interaction of IMA with HSA was monitored by FACE, however, fluorometric quenching and ultrafiltration studies demonstrated weak binding here. It seems, that electrophoretic conditions favor dissociation of IMA-protein adducts, however this observation would require more detailed investigations. Albumin binding of IMA was generally considered less important in clinical practice, however binding constants reported in the range of *K*′ = 10^4^–10^5^ are considerable (Table 3). Among these reports, Di Muzio et al. used fatty acid free HSA and applied rather high IMA concentrations for the fluorometric assays without correction of the intensities by inner filter effect [28]. Gambacorti-Passerini et al. carried out ultrafiltration experiments, however it is not clear how they accounted for adhesion to the filter [32]. This uncertainty also applies for their AGP binding constant [32]. The results of the present study suggest a stability constant about one order of magnitude lower for the HSA–IMA adduct, which is in relatively good agreement with the data reported by Fitos et al. [26].

To illustrate the effect of the uncertainty in the HSA–IMA binding constant, model calculations depicted in Figure 13 were carried out. Variation of total concentrations of IMA and AGP on the free levels of IMA in blood was modelled as well. Figure 13a shows well, that, albumin has considerable impact on the distribution of IMA only if the HSA–IMA binding constant is *K′* > 10^3^ and the concentration of AGP corresponds to the normal average level (15 μM) [36]. An average total peak concentration (*c*_max_) of IMA in blood serum is reported to be 2 µM, its free fraction slightly decreases from ca. 7% to 5% if the HSA–IMA constant *K*′ = 10^4^ (the upper limit of our fluorometric estimation) applies instead of *K*′ = 10^3^. The relevance of albumin binding further lessens with two- and three-fold elevated AGP levels (Figure 13b,c), and less than 4% of total IMA is unbound under these conditions. In all, the AGP binding of IMA is more important compared to HSA. Moreover, AGP has a relatively low normal plasma level, but in acute-phase response a two- or three-fold increase in its concentration results in reduction of free IMA level to one half or one third in comparison to the one observed under physiological conditions according to the model calculations (Figure 13). This finding explains why elevated AGP levels may prevent the efficacy of IMA [32,38].

## 3. Materials and Methods

### 3.1. Materials

The antiviral agents FAVI, IMA and MOLNU are products of MedChemExpress (Monmouth Junction, NJ, USA); WF, DG, DIP, D_2_O, HSA (A8763, essentially globulin free), AGP (G9885), transferrin (T3309, containing physiological amount of iron) and human blood serum (H4522, male AB plasma) were purchased from Sigma–Aldrich (Merck KGaA, Burlington, MA, USA). Inorganic compounds such as KCl, NaCl, KH_2_PO_4_ and Na_2_HPO_4_ × 2 H_2_O, boric acid, sodium trimethylsilylpropanesulfonate (DSS), 2-(*N*-morpholino)ethanesulfonic acid (MES) and 4-(2-hydroxyethyl)-1-piperazineethanesulfonic acid (HEPES) were products of Molar Chemicals (Halásztelek, Hungary) or Reanal (Budapest, Hungary) in puriss quality.

### 3.2. Stock Solutions and Sample Preparation

For preparation of stock solutions and samples, Milli-Q water was used. Stock solutions of FAVI and MOLNU were prepared in water or in phosphate buffered saline (PBS, pH = 7.40) in 1–10 mM concentration. Imatinib could be dissolved in 0.01 M hydrochloric acid (*c* = 10 mM). The HSA, AGP and transferrin stock solutions were prepared in PBS buffer (pH = 7.40). The concentration of HSA and AGP stock solutions was determined on the basis of the reported molar absorbances: *ε*(280 nm) = 36,850 M^−1^cm^−1^ *ε*(280 nm) = 24,140 M^−1^cm^−1^, respectively [65,66,67]. The concentration of transferrin stock solutions was determined based on the average molar weight reported by the producer and calculated molar absorbance at 280 nm fall between the reported values of the apo- and holoprotein [68]. Stock solutions of WF and DG were prepared as described previously [54]. Hydrolyzed MOLNU (h-MOLNU) solutions composed of EIDD-1931 and isobutyric acid were prepared by incubating MOLNU at alkaline conditions (pH ~ 12) for 2 h. All protein containing samples were prepared in PBS and incubated usually at room temperature or at 37 °C for some of the measurements.

### 3.3. pH-Potentiometry

pH-potentiometric titrations of the antiviral agents were carried out similarly to our former works [69,70]. Proton dissociation constants (*K*_a_) were determined for the compounds at *I* = 0.1 M KCl, 25 °C with the Hyperquad2013 software [71,72,73].

### 3.4. Thermodynamic Solubility (S)

The thermodynamic solubility of IMA was measured for the saturated solutions in water at various pH values (pH 2: 0.01 M HCl, pH 6: 20 mM MES) buffer), pH 7.4 and 8.2: 20 mM HEPES buffer) at 25.0 ± 0.1 °C. Solutions in 10 mM concentration were attempted to make. Samples were mixed overnight, then sedimentation of the solid fraction was forced by centrifugation (Eppendorf, MiniSpin Plus centrifuge, 15 min, 12 000 rpm). The concentration of the compounds was determined by UV-vis spectrophotometry using stock solutions of the compound with a known concentration dissolved in DMSO, 50% and 10% (*v*/*v*) DMSO/buffered aqueous solution for the calibration. In the case of FAVI and MOLNU 10 mM solutions at pH 2 and 7.4 could be prepared without the presence of non-dissolved compound.

### 3.5. Lipophilicity

Distribution coefficient (*D*_pH_) values of the compounds were determined by the traditional shake-flask method in *n*-octanol/buffered aqueous solution at various pH values at 25.0 ± 0.2 °C. The pH values of the samples were adjusted with *n*-octanol pre-saturated aqueous solutions containing different buffer systems in ca. 20 mM concentration and 0.10 M KCl. The buffer systems used were as follows: pH 2: 0.01 M HCl; pH 4–6: MES and its Na-salt; pH 6–8: NaH_2_PO_4_/Na_2_HPO_4_; pH 8–10: boric acid/borate. Compounds were dissolved in buffered aqueous solutions pre-saturated with *n*-octanol. The final compound concentration was 10−200 µM, and the pH values were precisely monitored. Stock solutions and water pre-saturated *n*-octanol were gently mixed in 1:1, 1:0.1 or 1:0.01 volume ratio for 2 h. After phase separation, the UV-vis spectrum of the compound in the aqueous phase was compared to that of the original stock solution and the *D*_pH_ values of the compounds were calculated according to the following equation:$$D_{\mathrm{pH}\;}=\left[\frac{{\mathrm{Abs}\;}_{(\mathrm{stock}.\;\mathrm{sol}.)}}{{\mathrm{Abs}}_{\;(\mathrm{aqueous}\;\mathrm{phase}\;\mathrm{after}\;\mathrm{separation})}}-1\right]\times \frac{V_{\;(\mathrm{aqueous}\;\mathrm{phase})}}{V_{\;(\mathrm{n-octanol})}}$$

An Agilent Cary 8454 diode array spectrophotometer (Agilent Technologies, Santa Clara, CA, USA) was used to measure the UV-vis spectra in the interval 200–500 nm. The pH dependency of the *D*_pH_ values was used to estimate the proton dissociation constants (*K*_a_) of FAVI and IMA, as stated in our previous work [74].

### 3.6. ^1^H NMR Spectroscopy

^1^H NMR spectroscopic measurements were carried out on a Bruker Avance III HD instrument (Billerica, MA, USA). The spectra were recorded with WATERGATE water suppression pulse sequence using DSS internal standard and 10% (*v*/*v*) D_2_O. Aqueous stability of MOLNU was followed at pH 2.0, 7.4 and 12.0. Interaction between MOLNU (1 mM) and HSA (630 μM), AGP (15 μM), transferrin (45 μM) or 2-fold diluted blood serum was studied in PBS buffer. Quantification of h-MOLNU in Appendix A was done as follows: the spectra containing MOLNU were subtracted by the spectrum of the serum, then the peak at ca. δ = 1.2 ppm corresponding to h-MOLNU was integrated, the saturation phase of the plot was aligned to 100%.

### 3.7. Spectrofluorometry

Steady-state fluorescence studies were implemented by a Fluoromax (Horiba Jobin Yvon, Longjumeau, France) fluorometer in 1 cm quartz cells. All samples contained 1–5 μM HSA or 0.5 μM AGP and various protein-to-antiviral agent ratios (up to ca. 60 μM compound concentrations) were used. Site marker displacement experiments were also carried out. In these setups, the HSA-to-site marker (WF or DG) ratio was 1:1 and the concentration of the antiviral compounds was varied. Instrumental settings are listed in Appendix A. The computer program PSEQUAD was used for calculation of binding constants (*K*′) for protein–antiviral compound adducts similar to the approach described in our former works [52,54,75]. Calculations were always based on data obtained from at least two independent measurements. Corrections for self-absorbance and inner filter effect were done as described in our former works using the formula suggested by Lakowicz [54,59,75].

Fluorescence anisotropy measurements were made for the AGP–DIP–drug systems (2 μM–1 μM–0–50 μM IMA, MOLNU or FAVI) and AGP–FAVI (5 μM to 0.5 μM) samples. The fluorescence anisotropy signal ® of DIP or FAVI was followed (see wavelength settings in Appendix A) by the use of an automated polarizer set mounted into the steady-state instrument. Emission intensity was measured at *HH*, *HV*, *VV*, and *VH* (*H* = horizontal, *V* = vertical) orientations of the polarizers (placed on the excitation and emission sides, respectively) repeatedly, until the relative standard deviation of *r* decreased below 2% or a maximum of 15 cycles were run. The *G*-factor was determined separately as well, *G* = *Int._HV_*/*Int._HH_* = 0.94. The FluorEssence (v.3.9) software automatically calculated the overall anisotropy according to the equation *r* = (*Int._VV_* − *Int._VH_* × *G*)/(*Int._VV_* + 2 × *Int._VH_* × *G*) [59]. The overall fluorescence anisotropy (*r̅* ) of a system is equal to the intensity weighted (*f*_i_) sum of the anisotropies (*r*_i_) of each component: *r̅* = Σ {(*r*_i_ × *f*_i_)/Σ *f*_i_}. Consequently, only those ternary systems can be interpreted simply, where (i) the fluorescence intensity of the marker is barely sensitive to the protein binding (*f*_free_ ≈ *f*_bound_) and (ii) the competitor together with (iii) the protein are not fluorescent under the applied conditions.

The fluorescence lifetime of FAVI was measured on the same fluorometer equipped with a DeltaHub TCSPC controller using NanoLED light source N-360 (Horiba Jobin Yvon). The resolution of the system was 25 ps. See details on the instrument settings in Appendix A. Ludox^®^ (from Sigma–Aldrich) was used as scatter solution to obtain the instrument response function (IRF). The background (obtained with blank samples) was subtracted from the decay. The program DAS6 (version 6.6.; Horiba, Jobin Yvon) was used for the analysis of the experimental fluorescence decays. The fluorescence intensity (*Int.*) decay over time is described by a sum of exponentials,
(1)$$Int.\left(t\right)=\overset{n}{\underset{i=1}{\sum }}\alpha_{i}exp\left(\frac{-t}{\tau_{i}}\right)$$
where α*_i_* and τ*_i_* are the normalized amplitude and lifetime of component *i* respectively. The quality of the fit was judged from a χ^2^_R_ value close to 1.0 (χ^2^_R_ ≤ 1.20) and a random distribution of weighted residuals.

### 3.8. UV-Visible Spectrophotometry

An Agilent Carry 8454 diode array spectrophotometer was used to obtain UV-vis spectra in the interval 190–1100 nm, the path length (*l*) was 1 cm. The spectrophotometric titrations were performed with samples containing 25–200 μM compound over the pH range 2.0–11.5 at an ionic strength of 0.10 M (KCl) and at 25 °C. Proton dissociation constants together with the individual molar absorbance spectra were calculated with the computer program PSEQUAD [52]. The program requires the following input data: measured spectra together with the corresponding pH and analytical (total) concentration of the compounds; species matrix, with approximate protonation constant(s) and the absorbing species need to be defined. The Newton–Raphson iterative method was used to obtain protonation constant(s) with standard deviations and molar spectra of the individual species.

### 3.9. Ultrafiltration

Samples were separated by ultrafiltration through 10 kDa membrane filters (Millipore, Amicon Ultra-0.5) into low and high molecular mass (LMM and HMM) fractions as described in our former work [54]. Samples contained various amounts of HSA (50–630 μM) or AGP (15–30 μM) and 25–200 μM antiviral agents. Blood serum was used without dilution or diluted to 2- or 4-fold with PBS. The concentration of the non-bound compounds in the LMM fractions was determined by UV-vis spectrophotometry by comparing the recorded spectra to those of reference samples without the protein. The LMM fraction of ultrafiltered blood serum possesses absorbance between 200 and 300 nm, which was taken into account during treatment of the spectra of samples containing the antiviral compounds and serum as well.

### 3.10. Capillary Electrophoresis

Frontal analysis capillary electrophoresis (FACE) measurements were performed on an Agilent 7100 capillary electrophoresis system Santa Clara, CA, USA) equipped with a diode-array detector (210–600 nm). For all experiments bare fused silica capillaries of 48 cm total length (50 μm inner diameter) were used (BGB Analytik, Boeckten, Switzerland). The background electrolyte (BGE) was PBS buffer (pH 7.40). The conditioning process of new capillaries and daily preparation were done as described formerly [55]. In order to ensure the steady baseline, the capillary was flushed with BGE (2 min) before each run and was rinsed with NaOH (0.1 M; 1.5 min), H_2_O (1.5 min), and then with BGE (2 min) after each separation. The sample tray and the capillary were kept at a constant temperature of 25 °C. Samples were injected hydrodynamically at 50 mbar for 30 s, and voltage of 10 kV was applied for the separation process producing a current of ca. 180 μA. Electropherograms were recorded and evaluated by the program ChemStation (Agilent Technologies). Protein-to-compound concentration ratio was between 0:1 and 5:1, the compound concentration was usually kept constant (*c*_IMA_ = 15 μM, *c*_FAVI_ = 30 μM), and concentration of HSA or AGP was varied. Samples containing IMA in various concentration (0–16 μM) and constant AGP concentrations (5.6 μM) were made as well. The concentration of the non-bound compound was calculated from plateau heights using external calibration.

### 3.11. Model Calculations

Model calculations were done with the software MEDUSA (32 bit version) [76] using log*K*′_HSA-IMA_ = 2.0–4.0; log*K*′_AGP–IMA_ = 6.0; c_IMA_ = 0.01–5 μM; c_HSA_ = 630 μM, and c_AGP_ = 15, 30 or 45 μM.

## 4. Conclusions

Serum protein binding of three approved antiviral drugs, FAVI, MOLNU and IMA, used to treat COVID-19 was investigated in this work. In addition to the protein binding assays, proton dissociation processes, aqueous solubility, stability, and lipophilicity at various pH values were also studied by means of pH-potentiometry, UV–visible and fluorescence spectroscopies, and/or by *n*-octanol/water partition experiments. The lipophilicity of the three drugs differs significantly and is highly dependent on the milieu, either they are in the gastric juice (a typical acidic pH: 2.0) or circulating in the blood at pH 7.4. A clear correlation is visible for IMA regarding its actual protonation state and lipophilicity. At pH 2.0, IMA is extremely hydrophilic (log*D*_2.0_ = −2.6) and exceedingly soluble in water (*S* > 10 mM), with the majority of the pH 2.0 present in the solution as +3 and +4 charged cations. At pH 7.4, the basic groups of IMA are primarily deprotonated, with a 3:1 ratio of the +1 charged form to the neutral species. Therefore, IMA is rather lipophilic at this pH (log*D*_7.4_ = +2.4) and possesses poor aqueous solubility (*S* = 24 μM, 37 °C). Molnupiravir and h-MOLNU are hydrophilic at both acidic and neutral pH due to the sugar moiety; it is charge neutral at the pH of the blood, while about half of it is already +1 charged at pH 2.0. A second proton dissociates at pH > 9, and the ester bond in MOLNU hydrolyzes at pH 12.0 rapidly resulting in the formation of EIDD-1931 and the isobutyrate ion. In contrast to the former two agents, FAVI is slightly lipophilic at pH 2.0 (log*D*_2.0_ = +0.15) and becomes very hydrophilic when it gets deprotonated and −1 charged above pH 4 (log*D*_7.4_ = −1.99). Both FAVI and MOLNU are well soluble in water between pH 2.0 and 8.2. The actual protonation states of the compounds provide a satisfactory explanation for the observed lipohilic-hydrophilic tendencies. Favipiravir is highly fluorescent in its deprotonated form in aqueous solution, but shows less fluorescence in ethanol, *n*-octanol, and water, at pH < 4. Fluorescence lifetime data were also determined in these media.

The distribution of the compounds in blood was investigated in whole serum and in binding experiments with selected transport proteins HSA and AGP by means of membrane ultrafiltration, frontal analysis capillary electrophoresis, steady-state fluorometry, and fluorescence anisotropy techniques. The ester bond in MOLNU is hydrolyzed rather slowly by the protein constituents of blood serum (takes about one day and the hydrolysis is most likely catalyzed by butyrylcholinesterase. This may be important if MOLNU was delivered intravenously into the vascular system. Molnupiravir and its hydrolyzed form do not bind considerably to the HMM fraction of blood serum. Favipiravir bound 18–28% to the protein fraction of blood serum in membrane ultrafiltration experiments but did not show any binding to HSA or AGP in ultrafiltration, capillary electrophoresis and fluorometric studies. This observation contradicts the only available literature data indicating 54% protein binding in vitro, where albumin was indicated as major carrier. To resolve this contradiction in the future, we recommend taking into account any ethnic disparities, in addition to rigorous analysis design. Imatinib strongly binds to AGP, the binding constant varies within log*K*′ = 5.8–6.0 depending slightly on the temperature (25 °C or 37 °C) or the storage mode of the protein. Fluorometric experiments revealed a much lower affinity for albumin (log*K*′ ≤ 4.0 at 25 °C and 37 °C). The FACE experiments indicated weaker binding of IMA towards AGP and no binding to HSA at 25 °C. This underestimation may result from the electrophoretic conditions (electric current, high voltage) or from the frontal analysis technique. We could model the blood distribution of IMA under physiological and pathological conditions using the binding data determined in this study. Model calculations for the IMA–AGP–HSA ternary system revealed that AGP levels play a more important role in the free fractions of IMA than HSA. Our model explains well the clinical observations that elevated AGP levels seemed to prevent the anticancer effect of IMA.

With the present work, we also aimed to point out that it is worthwhile to perform in vitro solution chemistry and biodistribution studies on approved drugs to interpret the phenomena already observed in vivo on patients (e.g., fluorescence of FAVI, possible inefficacy of IMA in the acute phase) and to complement the literature with this context.

## Data Availability

Authors can confirm that all relevant data are included in the article.

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
