# Peer review of "Evaluation of In Vitro Distribution and Plasma Protein Binding of Selected Antiviral Drugs (Favipiravir, Molnupiravir and Imatinib) against SARS-CoV-2"

_ijms, 2023, doi:10.3390/ijms24032849_

Round 1

Reviewer 1 Report

I appreciate the opportunity to review the manuscript entitled "In vitro blood distribution and plasma protein binding of selected antiviral drugs (favipiravir, molnupiravir, and imatinib) used against SARS-CoV-2." In the present study, the authors evaluated the in vitro distribution and plasma protein binding efficiency of recently repurposed drugs for the treatment of COVID-19 disease. However, the author may take note of the major and minor remarks listed below to improve the manuscript:

Major comments:

  1. As the complete study is based on the in vitro experiments with the mentioned blood components and plasma proteins, I would highly recommend changing the title to "Evaluation of in vitro distribution and binding efficiency of selected antiviral drugs against SARS-CoV-2."
  2. The conclusion in the abstract and manuscript is inadequate, and I would recommend rewriting.
  3. It is highly recommended to improve readability and conciseness; the manuscript requires extensive grammar checking and editing. For eg.
  • Line No. 34-35: The use of the phrase "slow and cumbersome development of new drugs"
  • Line No. 70: Use of the word "misleading"; instead, authors can write "contradicting."
  • Line No. 110-111: Please check the sentence, "Among the title compounds, MOLNU contains an ester functional group that possibly can hydrolyze under certain conditions." Instead, I would recommend writing "MOLNU, one of the title compounds, contains an ester functional group that may hydrolyze under certain conditions." to increase readability and conciseness.
  • Please make separate sections for the aqueous stability, proton dissociation processes, and lipophilicity in Section 2.

Minor comments:

  1. Please change Section 2 to "Results and Discussion."
  2. Please include "Section 3.6" in the methodology.

Author Response

Reply is found in the attachment.

Reviewer 2 Report

The manuscript entitled "In vitro blood distribution and plasma protein binding of se- 2 lected antiviral drugs (favipiravir, molnupiravir and imatinib) used against SARS-CoV-2" by Dömötör and Enyedy is nice and impressive piece of work. Even after couple of careful I could not find anything lacking in the manuscript. The methods, results, and discussions are well in accordance. My only suggestion for authors is to add future perspective mentioning authors' own take on how the findings of this manuscript is going to add value to relevant scientific research in near future.

Author Response

Please, find the answers in the attachment.

Reviewer 3 Report

The manuscript by Dömötör and Enyedy describes the investigation of acid-base and ADME properties of the antiviral drugs favipiravir, molnupiravir, and imatinib which were used for the treatment of COVID-19. Along with water solubilities, dissociation constants, and octanol-water distribution coefficients, the interaction of the compounds with blood serum proteins were studied by a series of relevant methods. Although the investigated chemical compounds are known, their above-mentioned properties are scarcely present in the literature, or contradictory data are available.  The results obtained by the authors are important and deserve publication.

Specific comments:

Table 1:  It is not clear what the digit "2" means which is given after the percent sign under "Species at pH 7.4".

The authors did not explain how the computed spectra shown in Figure 5 and Figure 1S were obtained. Methodological details of these computations should be included in the manuscript.

Summarizing, I recommend acceptance of the manuscript for publication after minor revision.

Author Response

(The authors gave the same response as above.)

Reviewer 4 Report

In the manuscript „In vitro blood distribution and plasma protein binding of selected antiviral drugs (favipiravir, molnupiravir and imatinib) used against SARS-CoV-2 by Dömötör and Enyedy the authors investigated the serum protein binding abilities and blood distribution behavior of the antiviral drugs Favipiravir (FAVI), Molnupiravir (MOLNU) and Imatinib (IMA). For this purpose, proton dissociation, aqueous solubility, stability and lipophilicity were analyzed by pH-potentiometry, UV-visible and fluorescence spectroscopies and n-octanol/water partition experiments and serum protein binding by FACE and different spectroscopy methodes. These data are important to give insights into drug absorption, distribution, elimination, toxicity and efficacy after administration.  

Major comments:

It makes total sense to analyze the chemical behavior of the different compounds using different biological relevant pH values, as mentioned in line 195-197. I do not understand why other conditions like the temperature were not chosen to simulate physiological conditions. Why is the temperature for most experiments 25 °C and not 37°C as it was used in table 3 and figure 11? I would assume that at least the thermodynamic solubility, the distribution coefficient and the lipophilicity might be temperature dependent. More importantly, it might be a critical factor for the blood serum protein binding and interaction with HSA and AGP studies.

Why did the authors test 50 µM of the compounds for the human serum interaction (line 265), 30 µM of hMOLNU (line 296) and why was the concentration of MOLNU 1 mM for the NMR spectroscopy (line 277)? It would be helpful if the authors could provide the reader with a short explanation.

Minor comments:

Line 118:  The other two compounds, FAVI and IMA were stable in aqueous media between pH 1 and 12.

The authors should show the data for FAVI and IMA, at least in the supplements.

Line 125: FAVI and MOLNU have excellent solubility (≥ 10 mM) at pH 2.0,

5.5, 7.4 and 8.2, while the solubility of IMA decreases with increasing pH (Figure 2).

The authors should show the data for FAVI and MOLNU, at least in the supplements.

Author Response

(The authors gave the same response as above.)

Reviewer 5 Report

REVIEWER'S REPORT

 Manucsript title: In vitro blood distribution and plasma protein binding of selected antiviral drugs (favipiravir, molnupiravir and imatinib) used against SARS-CoV-2 (Authors: Orsolya Dömötör and Éva A. Enyedy).

  It is, in my opinion, a fairly interesting work that has been experimentally carried out quite correctly, and after minor corrections, it can definitely be published in this journal. I advise making the following changes to the manusript text:

 In abstract, in lines 11-12, "...recently proposed as..." should be replaced by "...which have recently been proposed..."; in line 13, "...compounds..." could be replaced by "...substances...".

  In introduction (page 1), in lines 33-34, "...previously approved for other conditions emerged for as major strategy in the fight against the virus" should be replaced by "...previously licensed for other illnesses has emerged as a crucial strategy in the virus's combat."; in lines 39-41 "...FAVI and MOLNU (Scheme 1) are both nucleoside analog prodrugs originally developed for treatment of influenza disease; FAVI got approval in Japan in this indication." should be rewritten as "..FAVI and MOLNU (Scheme 1) are both nucleoside analog prodrugs that were originally designed for the treatment of influenza sickness; FAVI was approved for this application in Japan."; in lines 54-58, the sentence "Extremely low incidence of  COVID-19 infection was reported in CML patients treated with TKIs, and in vitro results suggested, that IMA as an inhibitor of Bcr-Abl tyrosine kinase can prevent the virus replication at the early stages of infection [11,13]. Immunomodulatory effect of IMA was also suggested being able to reduce extreme inflammatory response [12]." could be replaced by "In CML patients treated with TKIs, the incidence of COVID-19 infection was extremely low, and in vitro data revealed that IMA, as a Bcr-Abl tyrosine kinase inhibitor, can block viral multiplication in the early stages of infection. The immunomodulatory impact of IMA has also been claimed to be capable of reducing severe inflammatory responses [12]."; in lines 64-68 "All three drugs are administered orally, otherwise they differ profoundly from eachother in their pharmacokinetic behavior. Simple physico-chemical properties such as solubility, charge and lipophilicity together with the plasma protein binding is of high importance concerning drug absorption, distribution, elimination, toxicity or drug efficacy." could be rewritten as "All three medicines are delivered orally; otherwise, their pharmacokinetic behavior differs significantly. Simple physicochemical features like as solubility, charge, and lipophilicity, as well as plasma protein binding, are extremely important in terms of medication absorption, distribution, elimination, toxicity, and efficacy."; in lines 68 "...it is little known about these properties in the case..." could be replaced by "...there is little information available on these qualities in the cases..."; in lines 71-72 "It seems like this in vitro assayed protein binding data is referred later as an in vivo result in the literature [17-21]." should be written as "This in vitro protein binding data appears to be referred to in the literature [17–21] as an in vivo result."; in lines 96-97 "The available literature on the protonation processes, lipophilicity, thermodynamic aqueous solubility and blood serum protein binding of these compounds is rather incomplete." could be replaced by "The available literature on these compounds' protonation mechanisms, lipophilicity, thermodynamic aqueous solubility, and blood serum protein binding is fairly limited."; in lines 98-99 "...and to assess critically the literature data and also the possible shortages of techniques applied in this work." should be replaced by "...as well as to critically examine the literature data and potential shortcomings of the approaches used in this work."

   In Results (page 3), lines 126-129 the sentence "This is due to the gradual deprotonation of the ionized groups, consequently the good solubility obtained at pH 2.0 (S > 10 mM) drops to 28 μM at pH 8.2 where the neutral form predominates (vide infra)" should be paraphrased to "As a result of the gradual deprotonation of the charged groups, the good solubility attained at pH 2.0 (S > 10 mM) declines to 28 M by pH 8.2, where the neutral form predominates (vide infra)."; in lines 134-135 "The proton dissociation constants (pKa) were determined by pH-potentiometric, UV-visible (UV-vis) spectroscopic and fluorometric titrations and are collected in Table 1." should be repaced by "Table 1 summarizes the proton dissociation constants (pKa) measured using pH-potentiometric,  UV-visible (UV-vis) spectroscopic, and fluorometric titrations."; in line 145 "...namely..." replace by "...as..."; in line 149 "...was considered" replace by "...was taken into account..."; in lines 166-167 "The zwitterionic character is not feasible for this compound taking into account the trend of lipophilicity data determined at pH 2.0 and 7.4 (vide infra)." should be replaced by "Given the trend of lipophilicity data determined at pH 2.0 and 7.4, the zwitterionic character of this molecule is not plausible (vide infra)."; in line 188 "hindering the pH-potentiometric determination" should be replaced by " ... ...making pH-potentiometric difficult."; in lines 193-194 "Taking into account the uncertainty of both methods the match of these constants is acceptable." should be rewritten as "Given the uncertainty of both methods, the matching of these constants is acceptable."; in line 197-198 "At pH 7.4, the neural HL form is present for MOLNU, while FAVI is completely deprotonated (L–), and IMA is mainly positively charged, and only ca. 25% is neutral ???  under this condition." should be replaced by "At pH 7.4, MOLNU has the neutral HL form, whereas FAVI is entirely deprotonated (L-), and IMA is mostly positively charged, with just around 25% neutral."; in line 230 "Knowing that..." might be replaced by "Given that..."; in line 231 "...aromatic nature..." might be replaced by "...aromatic character..."; in line 289-249 "Our results confirm that FAVI (or its fluorescent metabolites) is most probably responsible for..."  should be written as "Our findings suggest that FAVI (or its fluorescent metabolites) is most likely responsible for..."; in lines "; in lines 247-250 " IMA and MOLNU (and h-MOLNU) displayed negligible fluorescence in aqueous solution at any protonation state both in ethanol and n-octanol. IMA possesses weak fluorescence in n-hexane (λEX = 287nm, λEM = 450 nm, Figure S7), while MOLNU is not fluorescent in this solvent." might be replaced by " In both ethanol and n-octanol, IMA and MOLNU (and h-MOLNU) fluorescence was minimal in aqueous solution at any protonation state. In n-hexane, IMA has weak fluorescence (EX = 287nm, EM = 450nm, Figure S7), whereas MOLNU is not fluorescent"; in lines 253-255 " In this section, our studies on the global serum protein binding, and interaction with HSA and AGP are presented in detail and the obtained results are discussed and compared with literature data in Section 2.4." might be rewritten as " Our experiments on global serum protein binding and interaction with 253 HSA and AGP are reported in detail in this section, and the acquired results are discussed and compared with literature data in Section 2.4."; in lines 265-266 " As a first step, the interaction of the three drugs (at 50 μM concentration) with whole human serum was studied by ultrafiltration." could be replaced by " In the first step, the interaction of the three medicines (at 50 µM concentration) with whole human serum was investigated using ultrafiltration."; in lines 267-268 " As Figure 6 shows the compounds display versatile behavior." should be rewritten as " As depicted in Figure 6, the compounds exhibit diverse behavior."; in lines 403-404 " Nearly no quenching can be observed in Figure 11 when AGP was titrated by FAVI, h-MOLNU or MOLNU." should be replaced by " Figure 11 shows almost little quenching when AGP was titrated with FAVI, h-MOLNU, or MOLNU."

  In Materials and Methods, in line 598 "The applied buffer systems were" could be replaced by " The buffer systems used were as follows..."; in line 601 "... and exact pH values were measured..." replace by "... and the pH values were precisely monitored..."; in lines 608-610 " pH 6 dependence of the DpH values were utilized to estimate proton dissociation constants (Ka) of FAVI and IMA as described in our former work [74]" replace by " The pH dependency of the DpH data was used to estimate the proton dissociation constants (Ka) of FAVI and IMA, as stated in our previous work [74]."

  In Conclusion (page 19). In lines 709-712 " IMA is especially hydrophilic (logD2.0 = –2.6) and very well soluble in water (S > 10 mM) at pH 2.0 being present mostly as +3 and +4 charged cations in the solution. The basic groups of IMA are mainly deprotonated at pH 7.4 and the ratio of the +1 charged form to the neutral species is 3:1." could be replaced by " At pH 2.0, IMA is extremely hydrophilic (logD2.0 = -2.6) and exceedingly soluble in water (S > 10 mM), with the majority of the pH 2.0 present in the solution as +3 and +4 charged cations.

At pH 7.4, the basic groups of IMA are primarily deprotonated, with a 3:1 ratio of the +1 charged form to the neutral species."; in lines 720-724 " The actual protonation states of the compounds provide a satisfactory explanation for the observed lipohilic-hydrophilic tendencies. FAVI is highly fluorescent in its deprotonated form in aqueous solution, but shows less fluorescence in ethanol, n-octanol and in water at pH < 4. Fluorescence lifetime data were determined as well in these media."; in lines 728-731 " The ester bond in MOLNU is hydrolyzed relatively slowly by the protein constituents of blood serum (takes ca. one day), most likely butyrylcholinesterase catalyzes the hydrolysis. This could have relevance if MOLNU would be administered intravenously into the circulatory system." should be replaced by " The ester bond in MOLNU is hydrolyzed rather slowly by the protein constituents of blood serum (takes about one day), and the hydrolysis is most likely catalyzed by butyrylcholinesterase. This may be important if MOLNU is delivered intravenously into the vascular system."; in lines 737-738 "... we propose consideration of possible ethnic or regional differences beside careful design of the analysis." replace by "... we recommend taking into account any ethnic disparities, in addition to rigorous analysis design."; in lines 740-741 " Much lower affinity was found towards albumin (logK’ ≤ 4.0) in fluorometric assays." may be rewritten as " Fluorometric experiments revealed a much lower affinity for albumin (logK' 4.0)."

    There are some questions about Figures 11 (page 11) and 12 (page 13), because the limits of Ccompound/CAGP, Ccompound/CHSA, and Ccompound/CAGP for each analyzed compound varied. Why were not the same limits of variation chosen for each tested substances? Theoretical (quantum mechanical etc.) calculations of the examined compounds in this work, attempting to uncover some parallels between experiments and theoretical predictions, would make the work substantially more appealing.   

Author Response

(The authors gave the same response as above.)

Round 2

Reviewer 4 Report

I would like to thank the authors for adressing my suggestions.